# Association of X-ray Absorptiometry Body Composition Measurements with Basic Anthropometrics and Mortality Hazard

**DOI:** 10.3390/ijerph18157927

**Published:** 2021-07-27

**Authors:** Nir Y. Krakauer, Jesse C. Krakauer

**Affiliations:** 1Department of Civil Engineering, City College of New York, New York, NY 10031, USA; 2Associated Physicians/Endocrinology, Berkley, MI 48072, USA; jckrakauer@gmail.com

**Keywords:** dual-energy X-ray absorptiometry, obesity, sarcopenia, risk assessment, anthropometry, body shape

## Abstract

Dual-energy X-ray absorptiometry (DEXA) is a non-invasive imaging modality that can estimate whole-body and regional composition in terms of fat, lean, and bone mass. We examined the ability of DEXA body composition measures (whole-body, trunk, and limb fat mass and fat-free mass) to predict mortality in conjunction with basic body measures (anthropometrics), expressed using body mass index (BMI) and a body shape index (ABSI). We used data from the 1999–2006 United States National Health and Nutrition Examination Survey (NHANES), with mortality follow-up to 2015. We found that all DEXA-measured masses were highly correlated with each other and with ABSI and that adjustment for BMI and ABSI reduced these dependencies. Whole-body composition did not substantially improve mortality prediction compared to basic anthropometrics alone, but regional composition did, with high trunk fat-free mass and low limb fat-free mass both associated with elevated mortality risk. These findings illustrate how DEXA body composition could guide health assessment in conjunction with the more widely employed simple anthropometrics.

## 1. Introduction

The basic anthropometrics of height, weight, and waist circumference (WC), and the derived quantities body mass index (BMI, weight adjusted for height) and a body shape index (ABSI, WC adjusted for weight and height) are robust predictors of mortality hazard as well as many health conditions and forms of morbidity [1,2]. However, it is recognized that these basic body measures do not directly measure body composition—in particular, the amount, location, and type of fat and muscle tissue [3,4,5,6]. number of imaging methods are available that provide such information, although all are much less widely used than basic anthropometrics, and their benefits for prognosis are less well quantified [7].

Of these imaging modalities, dual-energy X-ray absorptiometry (DEXA) scanning is considered a reference method for the assessment of human body composition, due to its relative non-invasiveness, good discrimination ability, and low cost compared to other techniques [8,9]. Originally employed only to monitor bone mass and density for fracture risk assessment, DEXA was established in the 1990s to also distinguish between fat and lean tissue in the different body regions [10,11]. DEXA measurements, along with other clinical and laboratory examinations, were conducted for the 1999–2006 cohorts of the United States National Health and Nutrition Examination Survey (NHANES), comprising over 10,000 individuals selected to be broadly representative of the national population. These NHANES measurements were used to develop age-specific body composition reference values [12,13].

With the availability of follow-up data on mortality of NHANES subjects, a number of studies have also assessed how body composition correlated with mortality. These studies focused on whether there was a contrast in the mortality association of fat mass versus fat-free mass [14,15,16], with trunk fat being possibly particularly associated with adverse cardiometabolic outcomes linked to abdominal obesity, and also whether a protective effect could be seen of high limb lean tissue mass (which is primarily muscle) [17,18,19,20]. However, while these studies mostly adjusted for BMI, they did not consider the correlation of DEXA body composition and its association with mortality risk with ABSI, which in some populations is a more powerful mortality predictor than BMI and which is correlated with fat percentage and with muscle mass [21,22,23,24].

The work presented here analyzes NHANES data to determine (a) the association between simple anthropometrics (BMI and ABSI) and DEXA-based whole-body and regional (limb, trunk) composition (fat and lean mass), and (b) the association between the DEXA-based measures and mortality hazard and the extent to which they improve hazard assessment over using only simple anthropometrics. Our goal is to contribute to defining the value of DEXA body composition scans for prognosis and to explore how these scans could be used in conjunction with the more widely employed simple anthropometrics.

## 2. Methods

### 2.1. NHANES

NHANES has been sampling the civilian non-institutionalized USA population since 1971 using a cluster approach. Some groups of public health interest (children, the elderly, black and Mexican-American people) were deliberately oversampled. We analyzed the 1999–2000, 2001–2003, 2003–2004, and 2005–2006 NHANES cohorts, for which body composition parameters were measured using DEXA. We considered all adults (age 18 and over) with complete basic anthropometric and DEXA measurements and mortality follow-up. Mortality outcomes for adult subjects were available from the National Center for Health Statistics through 2015 (9–16 years of follow-up, median: 12.5 years).

DEXA scans in NHANES used the QDR 4500A fan beam densitometer (Hologic, Inc., Bedford, MA, USA) and Hologic Discovery software version 12.1. People who were taller than 196 cm, heavier than 136 kg, or pregnant were not scanned. We also excluded individuals with any missing data due to limb amputations, prosthetics or implants, or other factors that affected scan quality (which were particularly common at higher BMI levels and older ages) [25]. Therefore, the cohort studied cannot be taken as representative of the entire adult population but only of those whose whole-body DEXA scans would yield complete and valid results.

The protocol for NHANES has been approved by the National Center for Health Statistics Research Ethics Review Board as consistent with the Declaration of Helsinki. Ethics approval was not needed for the current study because only anonymized, public-use data (https://www.cdc.gov/nchs/nhanes/index.htm, accessed on 1 June 2021) is employed.

### 2.2. Indices and Standardization

Anthropometric indices were calculated as follows [26,27]:(1)$$\mathrm{BMI}\equiv W\cdot H^{-2},$$
(2)$$\mathrm{ABSI}\equiv \mathrm{WC}\cdot H^{5/6}\cdot W^{-2/3},$$
where W designates weight, H height, and WC waist circumference.

Similar to BMI, the mass estimated by DEXA is divided by the square of measured height to construct body composition indices [28,29]. Thus, fat mass index (FMI) is total fat mass divided by $H^{2}$, fat-free mass index (FFMI) is total non-fat mass divided by $H^{2}$, and similarly for the trunk and limbs (limb fat mass index LFMI, limb fat-free mass index LFFMI, trunk fat mass index TFMI, trunk fat-free mass index TFFMI).

The anthropometric or body composition index values are converted to Z scores by subtracting the smoothed age and sex-specific mean and dividing by the standard deviation [23], thus following the general formula:(3)$$\mathrm{Z-score}\equiv \frac{\mathrm{value}-\mathrm{mean}}{\mathrm{std}.\mathrm{dev}.}.$$

To better understand the interrelation of basic anthropometrics and body composition, correlations were calculated between the Z scores of the different indices listed above. ABSI and BMI are defined so that the correlation between them is close to zero [27]. Following the same approach, allometrically adjusted forms of the body composition indices can be found that have close to zero correlation with BMI and ABSI by first finding the least-squares coefficients for the linear regression
(4)logindex∼intercept+αlogH+β,logBMI+γlogABSI+σsex
where sex is set to 1 for females and 0 for males, and then adjusting for anthropometrics accordingly:(5)$$\mathrm{adjusted}-\mathrm{index}=\frac{\mathrm{index}}{{\left(\frac{H}{\langle H\rangle }\right)}^{\alpha }{\left(\frac{\mathrm{BMI}}{\langle \mathrm{BMI}\rangle }\right)}^{\beta }{\left(\frac{\mathrm{ABSI}}{\langle \mathrm{ABSI}\rangle }\right)}^{\gamma }}.$$

Here, 〈H〉= 166 cm, 〈BMI〉= 26 $\mathrm{kg}\;m^{-2}$, and 〈ABSI〉= 0.0803 $m^{11/6}\;{\mathrm{kg}}^{-2/3}$ are median values from NHANES III, used to normalize the body composition measures to a standard height, weight, and WC. This makes the adjusted indices approximately independent of BMI and ABSI as well as height [23]. The adjusted indices are then transformed to Z scores, as before, which standardizes variability due to age and sex.

### 2.3. Statistical Modeling of Association with Mortality

Cox proportional hazard modeling [30] was used to assess the impact of body composition indices, with or without anthropometrics on death rate (mortality hazard) over the follow-up period, helping determine which aspects of body composition could be related to mortality risk and what forms these relationships took. All mortality hazard models were compared to a baseline model, which included only age (used as the timescale in the Cox model), sex (male/female), and race (black/nonblack) as predictors [23,27,31].

As in previous studies [23,32,33], the main measure of relative model performance was the Akaike information criterion (AIC), which was expressed as a difference from the baseline model:(6)$$\Delta_{\mathrm{model}\;m}\equiv {\mathrm{AIC}}_{\mathrm{baseline}}-{\mathrm{AIC}}_{\mathrm{model}\;m}.$$

Lower AIC (higher Δ) indicates models that perform better as mortality predictors for the sampled population. The expected likelihood of each model given the NHANES data is proportional to $e^{\Delta /2},$ so that a difference of 6 is the approximate threshold for significance at the 95% confidence level (since $e^{-6/2}\approx 5\%$) [34].

Two additional measures of model mortality-prediction performance were also computed and considered. One was the $R^{2}$ statistic, defined as a proportion of variation in mortality explained by the predictors of each model, so that higher $R^{2}$ suggests a model with greater explanatory power [35]. The other measure computed was concordance (*C*), defined as the fraction of pairs of individuals in the sample for which the one modeled to be at greater risk actually died sooner [36]. Concordance ranges from 0 to 1, with 0.5 the expected value for models with no skill and higher values indicating models that are more skillful at explaining variation in survival. Concordance is a generalization of the area under the receiver operating characteristic curve (AUC), which can also range from 0 to 1 and which can be interpreted as the fraction of pairs of individuals where the one who died had the higher predictor value. AUC in its basic form is not a suitable measure here because it does not straightforwardly account for the time to death and because it assumes the hazard to be a monotone function of the predictor, which we expected to generally not be the case [37,38].

The anthropometric and body composition Z scores, used as predictors, could enter into the mortality hazard models either linearly or nonlinearly. For a linear predictor, each unit increase in its Z score would increase or decrease the logarithm of mortality hazard by a constant amount. For a nonlinear predictor, the logarithm of mortality hazard would be shifted by an arbitrary smooth function of the Z score, expressed using a penalized spline basis, with corrected AIC being used to choose the function’s complexity [27,36,39]. Of the anthropometric and body composition predictors considered here, ABSI was entered into each model as a linear predictor, since it has previously been shown that the logarithm of mortality hazard has a near-linear dependence on the ABSI Z score [2,27], while BMI and the body composition indices were entered as nonlinear predictors to allow for the possibility of, for example, U-shaped associations with mortality risk.

All analyses were conducted in the R programming language, version 4.0.4 [40]. The implementation of Cox proportional hazard modeling in the coxph function of the survival package, version 3.2.7, was used [36,41].

## 3. Results

### 3.1. Sample Characteristics and Correlations

Basic demographics of the NHANES sample are given in Table 1. The average age is a slight underestimate because, to ensure privacy, those 85 years old and over were all listed as 85 in the released NHANES data and formed 1.5% of the sample. Adjusting for H, W, WC reduced the variance of body composition measures substantially, with the limb fat mass remaining with the highest coefficient of variation.

All the body composition indices showed very high positive correlations with BMI (ranging from 0.81 for LFFMI to 0.94 for FMI), since, as weight increases, the amount of mass in each compartment also increases, and also had high positive correlations with one another (upper right half of Figure 1). Some of the indices were also correlated with height, suggesting that these masses did not scale exactly with height${}^{2}$. Furthermore, ABSI correlated positively with TFMI (r=0.24) and negatively with LFFMI (r=−0.25), confirming that high WC for given height and weight is associated with relatively more trunk fat and less limb lean mass. As expected from its derivation based on allometric scaling theory [27], ABSI showed almost zero correlation with BMI (r=0.02).

After adjusting for anthropometrics, the correlations of body composition indices with BMI were greatly attenuated (|r|<0.08; lower left half of Figure 1). After adjustment, the fat-free mass indices were positively correlated with one another and negatively correlated with the fat mass indices, suggesting a differentiation between people with more fat throughout their body at a given height, weight, and waist circumference versus people with more lean tissue.

The regression coefficients in Table 2 (α,β,γ in Equation (Equation 4)) confirm the correlations seen between body composition indices and anthropometrics. Taller people have relatively less fat mass (particularly trunk fat) and more fat-free mass (particularly limb fat-free mass) at a given BMI. The different indices are not proportional to BMI (which would correspond to β=1) but rather show power-law scaling: as BMI increases, fat mass increases much faster than linearly, while lean mass increases slower than linearly. Finally, at a given height and BMI, high ABSI is correlated with relatively more fat, particularly trunk fat, and less limb fat-free mass. Overall, as indicated by the coefficients of determination $R^{2}$, over 90% of the variance in DEXA-measured adult whole-body composition indices, and over 85% of the variance in regional (trunk and limb) composition, is predictable from sex and basic anthropometrics (Table 2).

### 3.2. Associations with Mortality Hazard

First, we considered each anthropometric or body composition index as a single mortality predictor added to the baseline model with age, sex, and race. The performance of each of these is shown in Table 3 and Figure 2. ABSI was the best single mortality predictor, with significantly higher Δ than models with any of the other indices; log mortality hazard rose linearly as ABSI Z score increased (Figure 2b). BMI showed increased risk at below-average values but little sign of increased risk at above-average values (Figure 2a). This is different from earlier studies with NHANES populations, which found a U-shaped association between BMI and mortality, with significantly increased risk also at the high end of the BMI range [23,27]. The difference is likely due to people with morbid obesity being underrepresented in the subpopulation with valid DEXA scans, which was studied here.

The profiles for mortality risk as a function of both FMI and FFMI (Figure 2c), as well as TFFMI, LFMI, and LFFMI (not shown) were very similar to each other and to that for BMI (Figure 2a), consistent with their very high positive correlations. They all featured increased risk at low values, and lowest risk at near- and above-mean values. TFMI showed somewhat different behavior, with increased risk at values both substantially below and substantially above the population mean (Figure 2d).

Given that ABSI outperformed all the body composition indices as a linear predictor, we next investigated whether adding adjusted body composition indices to a model that also contained ABSI and BMI as predictors resulted in improved prediction over only using BMI and ABSI. These results are shown in Table 4 and Figure 3. As found in previous studies, the combination of BMI and ABSI results in a powerful predictive model for mortality. Their statistical independence helps explain why the increases in Δ, $R^{2}$, *C* in the combined model over baseline were very similar to the sum of the incremental increases for the models with just BMI and just ABSI. Adding body composition indices to this model resulted in changes in Δ that were either not statistically significant (for FMI and LFMI), suggesting that these indices offered no usable mortality prediction improvement over simple anthropometrics, or only marginally statistically significant (for FFMI and TFMI). The two exceptions were TFFMI and LFFMI, which both substantially improved mortality prediction when added to BMI and ABSI. Above-average trunk lean mass (high adjusted TFFMI) was associated with increased mortality hazard, as was below-average limb lean mass (low adjusted LFFMI) (Figure 3). Because adjusted TFFMI and LFFMI bore a positive correlation, these opposite tendencies were better expressed when both were added to the prediction model simultaneously, leading to the combined model (last line in Table 4) showing even more improvement than the sum of that incrementally found in the models that included TFFMI and LFFMI individually.

## 4. Discussion

DEXA body composition has been rated the gold standard for reasons of precision and low radiation exposure for inferring composition across the three tissue types (bone, lean, fat) from total body or regional (trunk, limbs) areal projections [8]. Over the last 30 years, DEXA densitometers have become a part of routine screening, assessment and monitoring of osteoporosis. Many of these same instruments can perform total body scanning and thus provide widespread but unrealized access to body composition measurements. Factors for underutilization have included lack of insurance coverage and normative values. The latter concern has been successfully addressed, largely by the NHANES population data. However, only DEXA bone densitometry is a covered service and performed routinely. Moreover, the low utilization may be most attributable to the failure to establish the clinical utility of DEXA body composition for diagnostic or risk assessment [42].

BMI and ABSI are allometric anthropometric indices that respectively correct body weight for height and waist circumference for BMI. ABSI was derived to meet allometric criteria of statistical independence from height and BMI [43,44]. In this study, we extended the adjustment of DEXA mass measurements beyond height and BMI to include ABSI. In fact, we found that a power-law dependence on height, BMI, and ABSI can estimate the whole-body DEXA body composition to rather high accuracy, comparable to that reported for other methods for measuring fat mass such as bioelectrical impedance analysis [45]. Anthropometric corrected DEXA body composition values will be independent of BMI and ABSI and may therefore better identify mortality associations with body composition that cannot be inferred from BMI and ABSI alone.

While body size and body composition are conceptually distinct, they are also closely connected. Obesity, as categorized by BMI, is usually associated with relatively more fat tissue, particularly in the trunk, as compared to lean tissue. WC is mostly predictable from height, weight, age, and sex [46], but variability in standardized WC, expressible using ABSI, has considerable diagnostic value, and also correlates with body composition, with higher ABSI associated with more trunk fat and less limb lean tissue. The ratio of height to WC has been used as a proxy for relative fat mass, with threshold values for elevated mortality risk determined using NHANES data [15].

After adjusting for BMI and ABSI, both DEXA measured total fat mass and trunk fat mass showed little correlation with mortality risk. This may reflect the acknowledged inability of conventional DEXA to isolate the most harmful fat depots, such as abdominal visceral fat, as compared to other trunk depots such as superficial subcutaneous fat that are not particularly harmful [7,47,48,49]. Our finding is consistent with [14], who found that in NHANES 1999–2006 trunk fat percentage was not significantly associated with mortality risk, although it was associated with cardiovascular disease mortality.

On the other hand, above-average adjusted trunk fat-free mass was associated with higher mortality risk, a linkage that, to our knowledge, has not been previously reported for DEXA body composition. One potential mechanism is that excess trunk lean tissue may tend to correspond to enlarged internal organs (organomegaly). For example, enlarged hearts correlate to worse prognosis of patients with cardiac disease [50]. In patients with autosomal dominant polycystic kidney disease, enlarged livers and kidneys indicated a much higher risk of malnutrition [51]. In newly diagnosed patients with symptomatic Waldenstrom macroglobulinemia, enlarged livers and spleens were associated with worse survival [52]. Mice with targeted expression of an Igf2 transgene in smooth muscle cells showed enlarged hearts and shortened lifespan, which was discussed as potentially relevant to the progression of human cardiac diseases [53].

While adjusted trunk fat-free mass was associated with higher mortality, adjusted limb fat-free mass was associated with lower mortality. This is consistent with low muscle mass being a risk factor for death and disability. About 75% of skeletal muscle mass is in the limbs, and therefore limb measurements using DEXA “can be considered as the reference standard for measuring muscle mass” [54]. Mortality associations using total body composition will be confounded as total body lean tissue includes visceral organs, body fluids and soft tissue calcifications and is only about 50% muscle [54].

Our findings with regard to trunk and limb fat-free mass could also be explored in relation to the capacity-load model of disease risk [55]. In this model, metabolic capacity is lower for people with low birthweight or childhood stunting, as evinced by less limb muscle mass and smaller internal organs. Metabolic load is imposed by stressors such as unhealthy food and exposure to pathogens and toxic pollutants. Lower metabolic capacity reduces the ability to tolerate metabolic load and hence increases vulnerability to disease in later life.

One limitation of our study is that obese people were disproportionately likely not to have valid DEXA scans, which would have led them to be excluded from our analyses. The limitation of DEXA scanners in imaging people with obesity [10] has been mitigated in recent years with newer detectors and software, along with half-body scans for those too large to be completely scanned [5]. To apply the body composition normals and risk profiles inferred from NHANES also requires attention to the properties of different DEXA machines and software.

Another limitation of the present work is that we considered associations in the entire NHANES population with valid data and did not study whether they vary between subgroups defined by factors such as sex, age, or ethnicity. Gene variants associated with high ABSI, for example, were found to differ between women and men [56]. Follow-up work could address this, although inference for subgroups will be limited by smaller sample sizes.

Sarcopenia has come to complement abdominal obesity as a recognized marker of ill health, especially in the elderly, and has been extensively studied [57,58,59]. Most studies involving anthropometrics or body composition modalities focus on the most conceptually associated measurements. Appendicular muscle (limb lean) mass is used in the definition of sarcopenia, while trunk fat mass is considered an indicator of abdominal obesity and visceral fat. In both NHANES and in the South Korean population, ABSI was found to be a good indicator of the sarcopenia criterion of DEXA-measured low limb lean mass among people with above-threshold waist circumference [60]. However, functional testing of strength and mobility is likely superior to muscle mass for assessing morbidity and mortality attributable to sarcopenia [61,62].

## 5. Conclusions

We report that DEXA-derived whole-body composition (fat or fat-free mass indices), both before and after anthropometric adjustment, adds little skill to mortality prediction compared to anthropometrics alone. For regional DEXA body composition, limb lean and trunk lean do seem to improve on just BMI and ABSI. Thus, based on the present analysis from NHANES data, mortality prediction using DEXA body composition indices may offer little advantage for mortality prediction over basic anthropometrics. WWAnthropometrics-adjusted DEXA, however, suggests added value from some body regions. The association between adjusted limb fat-free mass and mortality is consistent with the well-known association of sarcopenia with low appendicular muscle mass. The predictive value of adjusted trunk fat-free mass is a novel finding and perhaps reflects the adverse mortality implication of organomegaly.

## Figures and Tables

**Figure 1 ijerph-18-07927-f001:**
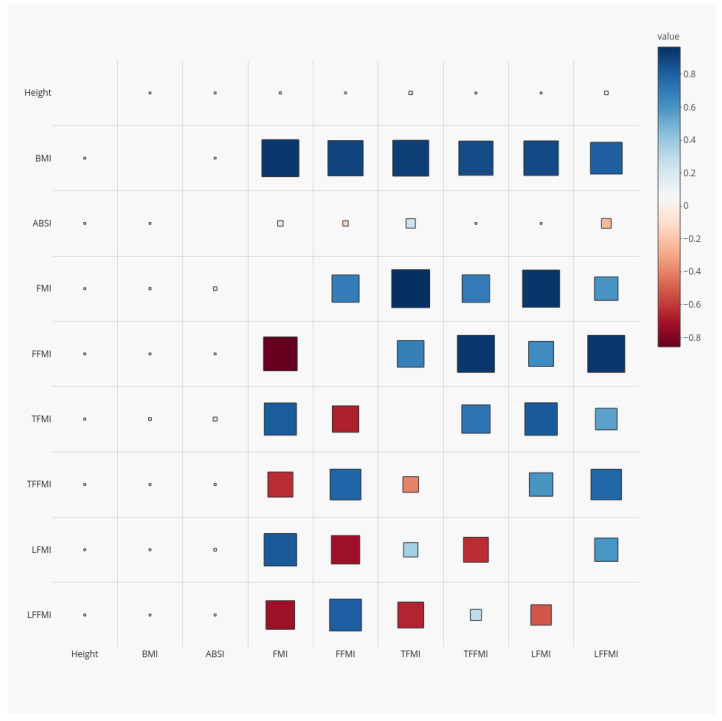
Correlation coefficients for anthropometrics and DEXA-derived body composition indices (Z scores relative to age- and sex-specific means) among NHANES 1999–2006 adults. Red squares denote positive correlations and blue negative ones; larger size indicates larger correlation magnitude. The upper right half of the correlation matrix shows correlations of the unadjusted indices, while for the lower left half, the body composition indices were adjusted to a standard height, weight, and waist circumference using power law scaling. NHANES = National Health and Nutrition Examination Survey; DEXA = dual-energy X-ray absorptiometry; BMI = body mass index; WC = waist circumference; ABSI = a body shape index; FMI = fat mass index; FFMI = fat-free mass index; TFMI = trunk fat mass index; TFFMI = trunk fat-free mass index; LFMI = limb fat mass index; LFFMI = limb fat-free mass index.

**Figure 2 ijerph-18-07927-f002:**
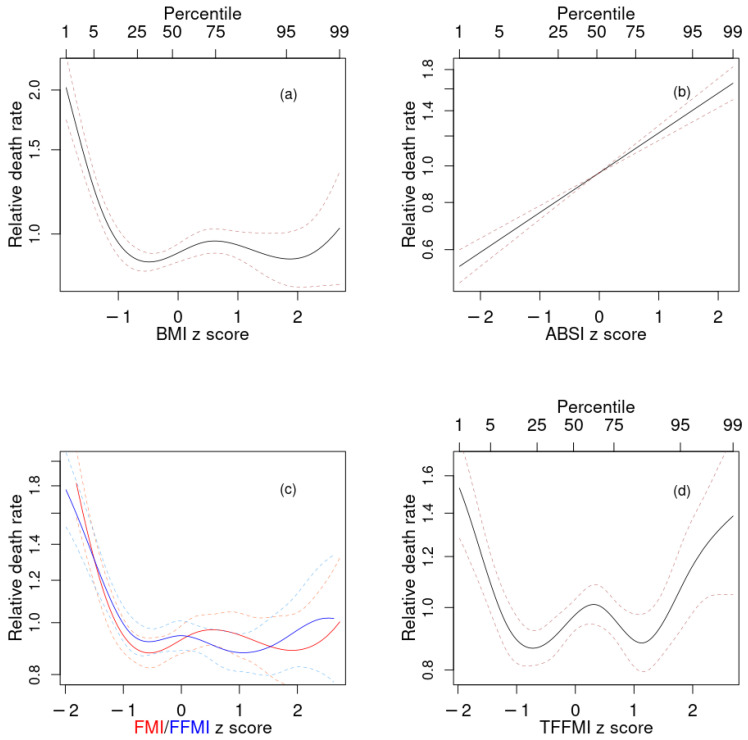
Estimated mortality hazard ratios in NHANES 1999–2006 as nonlinear (penalized spline) functions of (**a**) BMI, (**b**) ABSI, (**c**) FMI (red) and FFMI (blue), (**d**) TFFMI. Dashed lines indicate 95% confidence intervals. NHANES = National Health and Nutrition Examination Survey; BMI = body mass index; ABSI = a body shape index; FMI = fat mass index; FFMI = fat-free mass index; TFFMI = trunk fat-free mass index.

**Figure 3 ijerph-18-07927-f003:**
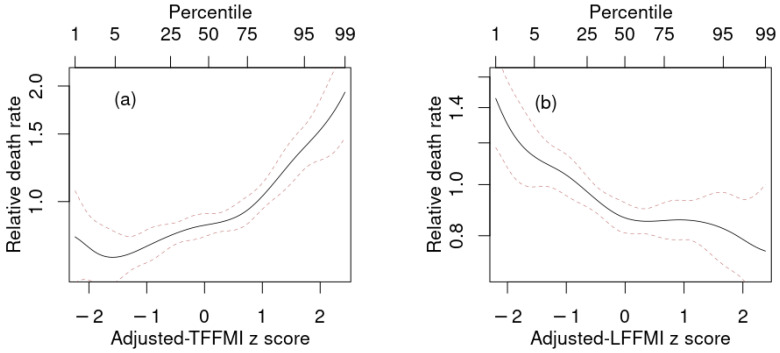
Estimated mortality hazard ratios in NHANES 1999–2006 as nonlinear (penalized spline) functions of adjusted (**a**) TFFMI, (**b**) LFFMI in models that also include as predictors BMI and ABSI. Dashed lines indicate 95% confidence intervals. NHANES = National Health and Nutrition Examination Survey; BMI = body mass index; ABSI = a body shape index; FMI = fat mass index; TFFMI = trunk fat-free mass index; LFFMI = limb fat-free mass index.

**Table 1 ijerph-18-07927-t001:** NHANES 1999–2006 sample characteristics.

	Valid DEXA Scans	All Adults
Number	14,064	19,959
Deaths	2140	3478
% female	48	52
Ethnicity	Mexican: 25%	24%
	Other Hispanic: 4%	4%
	White: 47%	47%
	Black: 20%	21%
	Other: 4%	4%
Age (y)	43 ± 19	46 ± 20
Height (cm)	168 ± 10	167 ± 10
Weight (kg)	76 ± 17	79 ± 20
BMI ($\mathrm{kg}\;m^{-2}$)	27.1 ± 5.1	28.1 ± 6.3
WC (cm)	94 ± 14	96 ± 16
ABSI ($10^{-2}\;m^{11/6}\;{\mathrm{kg}}^{-2/3}$)	8.04 ± 0.52	8.10 ± 0.54
FMI ($\mathrm{kg}\;m^{-2}$)	9.2 ± 3.8 [8.6 ± 2.0]	
FFMI ($\mathrm{kg}\;m^{-2}$)	18.1 ± 2.8 [17.9 ± 1.8]	
TFMI ($\mathrm{kg}\;m^{-2}$)	4.5 ± 2.1 [4.1 ± 0.9]	
TFFMI ($\mathrm{kg}\;m^{-2}$)	8.8 ± 1.3 [8.8 ± 0.9]	
LFMI ($\mathrm{kg}\;m^{-2}$)	4.3 ± 1.9 [4.1 ± 1.3]	
LFFMI ($\mathrm{kg}\;m^{-2}$)	8.0 ± 1.5 [7.8 ± 1.0]	

Basic demographics and body measurements (mean ± standard deviation) for adults in the NHANES 1999–2006 cohorts, both for those with valid DEXA scans (studied here) and for the entire cohort. Quantities in square brackets are after adjustment to a standard height, weight, and waist circumference. NHANES = National Health and Nutrition Examination Survey; DEXA = dual-energy X-ray absorptiometry; BMI = body mass index; WC = waist circumference; ABSI = a body shape index; FMI = fat mass index (from DEXA scan); FFMI = fat-free mass index; TFMI = trunk fat mass index; TFFMI = trunk fat-free mass index; LFMI = limb fat mass index; LFFMI = limb fat-free mass index.

**Table 2 ijerph-18-07927-t002:** Regression coefficients for body composition indices from DEXA versus simple anthropometric indices.

Index	Height	BMI	ABSI	$R^{2}$
FMI	−0.362	1.830	1.246	0.930
FFMI	0.140	0.611	−0.364	0.912
TFMI	−0.867	2.183	2.354	0.918
TFFMI	0.013	0.600	−0.011	0.863
LFMI	0.202	1.666	0.423	0.882
LFFMI	0.491	0.672	−0.764	0.879

Regression coefficients for body composition indices versus the anthropometrics height, BMI, and ABSI (Equation (Equation 4)), plus the regression coefficient of determination $R^{2}$. BMI = body mass index; WC = waist circumference; ABSI = a body shape index; FMI = fat mass index; FFMI = fat-free mass index; TFMI = trunk fat mass index; TFFMI = trunk fat-free mass index; LFMI = limb fat mass index; LFFMI = limb fat-free mass index.

**Table 3 ijerph-18-07927-t003:** The association of each body measure with mortality hazard.

Predictor	Δ	$R^{2}$	*C*
Baseline	0	0.031	0.567
BMI	79.3	0.056	0.581
ABSI	115.1	0.064	0.602
FMI	72.0	0.055	0.582
FFMI	46.8	0.047	0.585
TFMI	47.6	0.047	0.579
TFFMI	31.1	0.043	0.583
LFMI	80.3	0.057	0.586
LFFMI	99.7	0.061	0.598

Results of Cox proportional hazard modeling for mortality risk in NHANES 1999–2006 with anthropometric or DEXA body composition index z scores taken as predictors. The baseline model only included as predictors age, sex, and race, which the other models also all included. NHANES = National Health and Nutrition Examination Survey; DEXA = dual-energy X-ray absorptiometry; BMI = body mass index; WC = waist circumference; ABSI = a body shape index; FMI = fat mass index; FFMI = fat-free mass index; TFMI = trunk fat mass index; TFFMI = trunk fat-free mass index; LFMI = limb fat mass index; LFFMI = limb fat-free mass index; Δ = Akaike information criterion score reduction relative to the baseline model; $R^{2}$ = measure of explained variation; *C* = concordance.

**Table 4 ijerph-18-07927-t004:** The association of each body composition measure with mortality hazard when considered alongside simple anthropometrics.

Predictor	Δ	$R^{2}$	*C*
BMI + ABSI	195.2	0.088	0.615
+FMI	200.6	0.096	0.618
+FFMI	207.5	0.097	0.620
+TFMI	202.0	0.094	0.618
+TFFMI	255.6	0.110	0.627
+LFMI	200.9	0.097	0.620
+LFFMI	222.4	0.100	0.619
+TFFMI + LFFMI	317.6	0.130	0.635

As in Table 3, but for models including both BMI and ABSI as predictors, and with the body composition indices adjusted to remove correlation with BMI and ABSI. NHANES = National Health and Nutrition Examination Survey; BMI = body mass index; WC = waist circumference; ABSI = a body shape index; FMI = fat mass index; FFMI = fat-free mass index; TFMI = trunk fat mass index; TFFMI = trunk fat-free mass index; LFMI = limb fat mass index; LFFMI = limb fat-free mass index; Δ = Akaike information criterion score reduction relative to the baseline model; $R^{2}$ = measure of explained variation; *C* = concordance.

## Data Availability

The public-use NHANES data analyzed in this study are available at https://www.cdc.gov/nchs/nhanes/index.htm, accessed on 1 June 2021.

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
