# Peer review of "Association of X-ray Absorptiometry Body Composition Measurements with Basic Anthropometrics and Mortality Hazard"

_ijerph, 2021, doi:10.3390/ijerph18157927_

Round 1
Reviewer 1 Report
Line 30 – are you sure about the whole name of NHANES?
Line 41 – you have some doubts concerning reference 20?
Methods – lack of statistical methods
Tabele 1-4 – I am not convinced to such type of tables and descriptions, the reader might hesitate if the text below the table refers to the table or is a part of a main text
Author Response
> Line 30 – are you sure about the whole name of NHANES?
The name also has "Examination". We added it now.
> Line 41 – you have some doubts concerning reference 20?
Thanks for catching that, one of the references was not displayed properly. Fixed now.
> Methods – lack of statistical methods
The statistical modeling for association with mortality is described in subsection 2.3. We changed its title to make this clearer.
> Tabele 1-4 – I am not convinced to such type of tables and descriptions, the reader might hesitate if the text below the table refers to the table or is a part of a main text
We now typeset the table notes in smaller font so that they can be easily distinguished from the main text.
Reviewer 2 Report
anthropometrics and explored all measures with mortality hazard in US adults. The topic is of interest and it would meaningful addition in this field. However, several issues warrant consideration before publication.
Line 63-64: Authors claimed that the sample in this study did not represent due to missing values and other reasons although it used NHNAES dataset. I recommend presenting the basic characteristics without DEXA measurement for the sample population along with whole population in Table 1. In that way, readers can find the difference of this sample population.
Line 88: <W>=26kgm-2 -> Is <W> right? It seems BMI.
Table 1: Please put some lines including top and bottom. The present format does not look like Table. In addition, please put the symbol of footnote and add abbreviation of FMI, TFMI and so on. All Tables including the title need to be self-explanatory.
Line 130: The average age is a slight underestimate -> If so, please present age as categorical variables, n(%)
Figure 1: Please add explanation of color meaning and abbreviation at footnote.
Table 2: Please revise the Table format. The title also needs to be self-explanatory. I recommend that ‘Regression coefficients for body composition indices by DEXA versus the simple anthropometric indices’ are title and add abbreviation at footnote.
Table 3 & Table 4: 1) Please reformat Tables. 2) Please change the titles. Table 3 represented ‘The association of each body measures with mortality hazard in US adults from NHANES 1999-2006’, while Table 4 represented the association of multiple body measures with mortality. 3) Add abbreviation in each Tables.
Discussion: The discussion is generally well-written. However, please organize the last paragraph separately into ‘5. Conclusions’.
Author Response
> The topic is of interest and it would meaningful addition in this field. However, several issues warrant consideration before publication.
> Line 63-64: Authors claimed that the sample in this study did not represent due to missing values and other reasons although it used NHNAES dataset. I recommend presenting the basic characteristics without DEXA measurement for the sample population along with whole population in Table 1. In that way, readers can find the difference of this sample population.
We now added a column to give summary statistics for the entire study population.
> Line 88: <W>=26kgm-2 -> Is <W> right? It seems BMI.
Thanks for catching this, we corrected it to <BMI>.
> Table 1: Please put some lines including top and bottom. The present format does not look like Table. In addition, please put the symbol of footnote and add abbreviation of FMI, TFMI and so on. All Tables including the title need to be self-explanatory.
We added horizontal lines and explanations of all abbreviations to the tables.
> Line 130: The average age is a slight underestimate -> If so, please present age as categorical variables, n(%)
Only the ages of those older than 85 were truncated. Tabulating each age from 18-85 individually would be unwieldy and not very insightful.
> Figure 1: Please add explanation of color meaning and abbreviation at footnote.
We added a list of the abbrevitaions. We also added an explanation of the color scheme.
> Table 2: Please revise the Table format. The title also needs to be self-explanatory. I recommend that ‘Regression coefficients for body composition indices by DEXA versus the simple anthropometric indices’ are title and add abbreviation at footnote.
Done.
> Table 3 & Table 4: 1) Please reformat Tables. 2) Please change the titles. Table 3 represented ‘The association of each body measures with mortality hazard in US adults from NHANES 1999-2006’, while Table 4 represented the association of multiple body measures with mortality. 3) Add abbreviation in each Tables.
Done.
> Discussion: The discussion is generally well-written. However, please organize the last paragraph separately into ‘5. Conclusions’.
Done.
Reviewer 3 Report
Title: Association of X-ray absorptiometry body composition measurements with basic anthropometrics and mortality hazard
In 2.2, the indices and standardized W, H, WC must indicate the unit.
Why did the author choose FMI, FFMI, TFMI, TFFMI, LFMI, LFFMI as body composition indicators? There should be other body composition indicators. What is the significance of the body composition index chosen by the author? Please explain.
This research considered the influence of gender. However, in the statistics and description of data, such as table 1, table 2, figure 1, there was no other description of male and female. It is recommended to supplement the analysis results and descriptions of male, female and mixed gender.
In this study, 2.3, "Association with Mortality" describes the indicators of mortality. The author should give a more specific description of the "mortality" in the sample.
Writing "statistical methods" of existing research papers is usually independent of a complete section. Why did the author add statistical methods to the sub-chapter section "2.3". The user tool was also not clear?
The existing NHANES research has been carried out until 2020. Why did the author only show the death rate quoted to 2015? And no recent valid information was used.
NHANES is a fairly complete investigation. In the past, has NHANES been used for related research? If so, what is the difference between this research and past research? In the manuscript, the author did not specifically explain or discuss it. In this study or in the manuscript, the author did not specifically mention similar research and progress. The author only used various indicators of NHANES' mortality and body composition for statistical analysis. If there are more special contributions or differences in this study, the authors are suggested to add more.
Author Response
> Title: Association of X-ray absorptiometry body composition measurements with basic anthropometrics and mortality hazard
> In 2.2, the indices and standardized W, H, WC must indicate the unit.
Conceptually any units could be used; this which simply change the units associated with the derived indices. We indicate the units used in each case whenever we present numerical values.
> Why did the author choose FMI, FFMI, TFMI, TFFMI, LFMI, LFFMI as body composition indicators? There should be other body composition indicators. What is the significance of the body composition index chosen by the author? Please explain.
There are certainly many possible body composition measures that could be considered. The ones we chose to study here are based on the previous work on the topic on which we build, as summarized in the Introduction. In general, previous studies have focused on whether fat mass was associated with differential health hazards compared to fat-free mass. In terms of body regions, the most attention has been paid to the trunk and limbs, in particular the potential roles of trunk fat and limb fat-free or lean tissue.
> This research considered the influence of gender. However, in the statistics and description of data, such as table 1, table 2, figure 1, there was no other description of male and female. It is recommended to supplement the analysis results and descriptions of male, female and mixed gender.
Considering that body composition is different on average between males and females, we normalized all anthropometric and body composition measures to sex (and age) specific z scores and looked at the impact of differences from these average patterns on mortality risk. In addition, all our Cox proportional hazard models included sex as an explanatory variable to allow for females experiencing lower mortality hazard than males on average. We now discuss as a direction for future research to also determine how the associations between between mortality hazard and body composition z score might vary by sex, or across other demographic subgroups such as age and ethnicity.
> In this study, 2.3, "Association with Mortality" describes the indicators of mortality. The author should give a more specific description of the "mortality" in the sample.
> Writing "statistical methods" of existing research papers is usually independent of a complete section. Why did the author add statistical methods to the sub-chapter section "2.3". The user tool was also not clear?
Section 2.3 describes the statistical modeling for association with mortality. We changed its title to make this clearer, and added references to the software used.
> The existing NHANES research has been carried out until 2020. Why did the author only show the death rate quoted to 2015? And no recent valid information was used.
We used the latest publicly available information of deaths, which as of now only extends to 2015.
> NHANES is a fairly complete investigation. In the past, has NHANES been used for related research? If so, what is the difference between this research and past research? In the manuscript, the author did not specifically explain or discuss it. In this study or in the manuscript, the author did not specifically mention similar research and progress. The author only used various indicators of NHANES' mortality and body composition for statistical analysis. If there are more special contributions or differences in this study, the authors are suggested to add more.
Indeed, NHANES has been used to study many health-related factors. In particular, several studies have previously considered mortality in followed-up NHANES subjects in relation to body composition. We summarize these studies and give full references in the second to last paragraph of the manuscript Introduction. However, we identified some gaps in this previous research, which led to the goals formulated for the current study and given in the last paragraph of the Introduction, namely "to determine (a) the association between simple anthropometrics (BMI and ABSI) and DEXA-based whole-body and regional (limb, trunk) composition (fat and lean mass), and (b) the association between the DEXA-based measures and mortality hazard and the extent to which they improve hazard assessment over using only simple anthropometrics".
Reviewer 4 Report
Dear authors, well done on a good piece of work. Though I think this study is of value and does contribute to the literature, I feel that the aim of the study and therefore some of the conclusions drawn do require some adjustment. Specifically, you have stated that your overall goal is to explore how DEXA scanning can be used in conjunction with simple, manual anthropometrics when performing routine health assessments. There are a few issues with this, first we know that DEXA can quantify body composition components which manual measures cannot, with several studies having correlated simple anthropometrics with body composition measured by DEXA. DEXA machines are still very expensive, though cheaper than MRI, they still cost in excess of $30,000 restricting their use to laboratory environments. Also, though they expose individuals to lower radiation exposure than CT, it is still a considerable amount of radiation in a short period of time. Typically in this field of research the aim is usually to develop lower cost, less invasive tools that identify those at risk as well as more expensive, invasive imaging modalities. Therefore, though I think this is a good study with valuable data and analysis, I think that the focus of the study should be adjusted to explore whether measures such as ABSI correlate with measures of body composition and whether they improve the assessment of health risk/mortality compared to commonly used measures such as BMI, WC, with DEXA used as the reference measure. The idea that DEXA scanning could be used in routine practice is perhaps unrealistic and the study should reflect that.
I have also provided a few specific comments below:
Line 1: Use of the phrase 'anthropometric measurements' is repetitive as the word anthropometrics itself means 'human measurement', simply stating anthropometrics is sufficient, which the author has done later in the paper.
Line 41: reference 20 is followed by a question mark, which is likely a typo.
Line 185: Interested in the finding that increased trunk lean mass was associated with increased mortality hazard. I note the discussion that this could be due to issues such as organomegaly, however, work by Jonathan Wells around the metabolic capacity-load model would argue that increased organ and muscle (lean) mass is indicative of greater metabolic capacity. Consideration of this could aid the discussion.
Line 216: 'Obesity as categorised by BMI is usually associated with more fat tissue, particularly in the trunk'. I would argue that this isn't strictly true, as it has been shown in previous studies that for a given BMI value there can be significant variations in waist girth and torso fat mass, due to the inability of BMI to differentiate between body tissue types within the body, or identify where mass is distributed.
Line 207: Bioelectrical impedance analysis does not directly measure fat mass, it measures body water and estimates the other body composition components. Greater specificity in this statement is needed.
Line 216: There are other adjusted anthropometric indices which could be assessed/compared against in this paper, such as the WHT.5R ratio developed by Nevill.
Line 220: 'Other trunk depots such as subcutaneous fat that are not particularly harmful'. I would question this, as there has been work by Dulloo et al. which has shown that the role of deep abdominal subcutaneous fat has been overlooked in previous research, finding that adverse metabolic effects (insulin resistance, dyslipidemia) are likely to result from both dysfunctional abdominal subcutaneous and visceral adiposity through the release of free-fatty acids. Clarification of this is necessary.
Author Response
> Dear authors, well done on a good piece of work. Though I think this study is of value and does contribute to the literature, I feel that the aim of the study and therefore some of the conclusions drawn do require some adjustment. Specifically, you have stated that your overall goal is to explore how DEXA scanning can be used in conjunction with simple, manual anthropometrics when performing routine health assessments. There are a few issues with this, first we know that DEXA can quantify body composition components which manual measures cannot, with several studies having correlated simple anthropometrics with body composition measured by DEXA. DEXA machines are still very expensive, though cheaper than MRI, they still cost in excess of $30,000 restricting their use to laboratory environments. Also, though they expose individuals to lower radiation exposure than CT, it is still a considerable amount of radiation in a short period of time. Typically in this field of research the aim is usually to develop lower cost, less invasive tools that identify those at risk as well as more expensive, invasive imaging modalities. Therefore, though I think this is a good study with valuable data and analysis, I think that the focus of the study should be adjusted to explore whether measures such as ABSI correlate with measures of body composition and whether they improve the assessment of health risk/mortality compared to commonly used measures such as BMI, WC, with DEXA used as the reference measure. The idea that DEXA scanning could be used in routine practice is perhaps unrealistic and the study should reflect that.
Thank you for raising these important points on the relative practicability of DEXA versus simple anthropometrics. We completely agree that DEXA is more expensive and cumbersome than simple anthropometrics, and also has a less favorable safety profile. Therefore, it is important to determine whether DEXA-based risk assessment actually outperforms simple anthropometrics, which could potentially justify the additional expense. Our results in addressing this question are mixed: we find that total-body composition from DEXA does not add much information to the risk assessment available from simple anthropometrics, but that some of the regional values (specifically limb fat-free and trunk fat-free mass) could contribute independent information. Whether this gain in information justifies routine DEXA scans will need to be studied further. We note that DEXA is already routinely used, at least in the USA, for bone density measurement to determine osteoperosis risk. If regional body composition could be obtained from the same scans, this may not add unreasonable monetary or safety costs.
> I have also provided a few specific comments below:
> Line 1: Use of the phrase 'anthropometric measurements' is repetitive as the word anthropometrics itself means 'human measurement', simply stating anthropometrics is sufficient, which the author has done later in the paper.
Good point. We rephrased.
> Line 41: reference 20 is followed by a question mark, which is likely a typo.
Thanks for catching that, one of the references was not displayed properly. Fixed now.
> Line 185: Interested in the finding that increased trunk lean mass was associated with increased mortality hazard. I note the discussion that this could be due to issues such as organomegaly, however, work by Jonathan Wells around the metabolic capacity-load model would argue that increased organ and muscle (lean) mass is indicative of greater metabolic capacity. Consideration of this could aid the discussion.
Thank you for this suggestion, we added this reference to our discussion.
> Line 216: 'Obesity as categorised by BMI is usually associated with more fat tissue, particularly in the trunk'. I would argue that this isn't strictly true, as it has been shown in previous studies that for a given BMI value there can be significant variations in waist girth and torso fat mass, due to the inability of BMI to differentiate between body tissue types within the body, or identify where mass is distributed.
We agree that this association is not strict, but it does hold statistically in the general population, cf. Table 2 where total fat mass scales with BMI to the 1.83 power, and trunk fat mass scales with an even higher power of BMI (2.18).
> Line 207: Bioelectrical impedance analysis does not directly measure fat mass, it measures body water and estimates the other body composition components. Greater specificity in this statement is needed.
We rephrased accordingly.
> Line 216: There are other adjusted anthropometric indices which could be assessed/compared against in this paper, such as the WHT.5R ratio developed by Nevill.
BMI and ABSI form a 'natural' set of simple anthropometric indices because they are statistically independent, whereas the waist-height ratio, for example, retains a high positive correlation with BMI. A full comparison of ABSI with waist-height ratio is carried out in the Krakauer and Krakauer 2014 cited reference.
> Line 220: 'Other trunk depots such as subcutaneous fat that are not particularly harmful'. I would question this, as there has been work by Dulloo et al. which has shown that the role of deep abdominal subcutaneous fat has been overlooked in previous research, finding that adverse metabolic effects (insulin resistance, dyslipidemia) are likely to result from both dysfunctional abdominal subcutaneous and visceral adiposity through the release of free-fatty acids. Clarification of this is necessary.
Thanks for pointing this out. We now differentiate superficial from deep subcutaneous abdominal fat, and have added more references on the topic.